# How Did the Two Years of the COVID-19 Pandemic Affect the Outcomes of the Patients with Inflammatory Rheumatic Diseases in Lithuania?

**DOI:** 10.3390/medicina59020311

**Published:** 2023-02-08

**Authors:** Jolanta Dadonienė, Gabija Jasionytė, Julija Mironova, Karolina Staškuvienė, Dalia Miltinienė

**Affiliations:** 1State Research Institute Centre for Innovative Medicine, LT-08406 Vilnius, Lithuania; 2Department of Public Health, Institute of Health Sciences, Faculty of Medicine, Vilnius University, LT-03101 Vilnius, Lithuania; 3Clinic of Rheumatology, Orthopedics Traumatology and Reconstructive Surgery, Institute of Clinical Medicine, Faculty of Medicine, Vilnius University, LT-03101 Vilnius, Lithuania

**Keywords:** rheumatic diseases, standardized mortality ratio, COVID-19, excess mortality

## Abstract

*Background and objectives*: the COVID-19 pandemic globally caused more than 18 million deaths over the period of 2020–2021. Although inflammatory rheumatic diseases (RD) are generally associated with premature mortality, it is not yet clear whether RD patients are at a greater risk for COVID-19-related mortality. The aim of our study was to evaluate mortality and causes of death in a retrospective inflammatory RD patient cohort during the COVID-19 pandemic years. *Methods*: We identified patients with a first-time diagnosis of inflammatory RD and followed them up during the pandemic years of 2020–2021. Death rates, and sex- and age-standardized mortality ratios (SMRs) were calculated for the prepandemic and pandemic periods. *Results*: We obtained data from 11,636 patients that had been newly diagnosed with inflammatory RD and followed up until the end of 2021 or their death. The mean duration of the follow-up was 5.5 years. In total, 1531 deaths occurred between 2013 and 2021. The prevailing causes of death in the prepandemic period were cardiovascular diseases, neoplasms, and diseases of the respiratory system. In the pandemic years, cardiovascular diseases and neoplasms remained the two most common causes of death, with COVID-19 in third place. The SMR of the total RD cohort was 0.83. This trend was observed in rheumatoid arthritis and spondyloarthropathy patients. The SMR in the group of connective-tissue diseases and vasculitis was higher at 0.93, but did not differ from that of the general population. The excess of deaths in the RD cohort during the pandemic period was negative (−27.2%), meaning that RD patients endured the pandemic period better than the general population did. *Conclusions*: The COVID-19 pandemic did not influence the mortality of RD patients. Strict lockdown measures, social distancing, and early vaccination were the main factors that resulted in reduced mortality in this cohort during the pandemic years.

## 1. Introduction

Inflammatory rheumatic diseases (RDs) are generally associated with premature mortality, mainly due to infections, premature atherosclerosis and subsequent cardiovascular complications, and major organ damage as a result of the RDs themselves [1,2,3,4,5,6]. 

Epidemiologic studies conducted before 2020 calculated that patients with inflammatory RD had up to 425% higher risk of death than that of the age- and sex-matched general population. The highest mortality was observed in patients with systemic connective-tissue diseases (CTDs; systemic sclerosis, systemic lupus erythematosus), systemic vasculitis, and myositis. The lowest mortality, comparable to the general population results, was observed in rheumatoid arthritis (RA), spondyloarthropathy (SPA), Sjogren’s syndrome, and polymyalgia rheumatica patient groups [7]. The greatest decrease in life expectancy at the time of birth in comparison with the general population was estimated in systemic sclerosis patients and is up to 34 years of life [1,8,9,10]. 

At the end of 2019, coronavirus disease 2019 (COVID-19), caused by the SARS-CoV-2 virus, emerged and spread fulminantly worldwide as a severe pandemic that was responsible for increased global mortality. According to the World Health Organization (WHO), COVID-19 has already caused over 6.67 million deaths [11]. However, the full weight of the COVID-19 pandemic is much greater than what is indicated only by reported deaths due to COVID-19. The full impact of the pandemic is assessed by calculating excess mortality, which is defined as the increase in all-cause mortality over the expected mortality rates. It is estimated that more than 18 million people died globally because of the COVID-19 pandemic (as measured via excess mortality) over the period between 1 January 2020 and 31 December 2021 [12]. The extent of excess mortality varied significantly between countries. The highest excess mortality rate due to COVID-19 was observed in Bolivia, which reached 734.9 deaths per 100,000 individuals. Negative excess-mortality rates (pandemic mortality was lower than that in the prepandemic period) were estimated in Iceland, Australia, Singapore, New Zealand, and Taiwan [12,13]. Older age and concomitant diseases such as dementia, chronic kidney disease, severe mental illness, cardiovascular disease, diabetes, chronic obstructive pulmonary disease, and cancer are predictors for severity and death related to COVID-19 [14]. However, it is still unclear whether patients with inflammatory RD are at a greater risk of COVID-19-related mortality. 

The objective of our study was to assess mortality and causes of death in a retrospective cohort of patients with inflammatory RD during the COVID-19 pandemic, in particular during 2020–2021, and to compare them with those of the general population of Lithuania and the results of the same cohort during the prepandemic years [7].

## 2. Materials and Methods

### 2.1. Data Sources

The retrospective cohort study was performed with data retrieved from Lithuanian compulsory health insurance information system database SVEIDRA. This is a population-based database that has been running since 1995 and registers all patients’ visits to healthcare institutions, the established diagnoses, and the prescriptions of state-reimbursed medications to all residents of Lithuania. The Vilnius Regional Bioethics Committee approved assessing these data and conducting the study (approval number: 158200-17-958-462, approval date: 7 November 2017).

Information about all patients who had been newly diagnosed with inflammatory RD during the period between 1 January 2012 and 31 December 2019 was obtained from SVEIDRA. Patients with RA, SPA (psoriatic arthritis (PsA), ankylosing spondylitis, and undifferentiated SPA), systemic CTD, and vasculitis were included. Information about the prescription of state-reimbursed medication such as glucocorticoids, conventional synthetic disease-modifying antirheumatic drugs (csDMARDs) (hydrochloroquine, sulfasalazine, methotrexate, azathioprine, leflunomide), or biological disease-modifying antirheumatic drugs (bDMARDs) (etanercept, adalimumab, infliximab, tocilizumab or rituximab with available biosimilars) was used for the verification of RD cases.

A total of 95,289 RD cases with a first-time diagnosis of inflammatory RD established between 2012 and 2019 were selected. A case was considered first-time-diagnosed if the patient had had at least 1 year of no previous RD record in the database. As no data preceding 2012 were available, we excluded 22,526 first-time diagnoses in 2012 from the total cohort, as it was not possible to verify their RD diagnosis prior to the index date.

We excluded 2251 cases from the cohort because they were younger than 18 years old at the time of diagnosis, and 10 because of an unidentifiable identification code. To verify the cases, we excluded 58,866 patients with no records of at least one prescription of state-reimbursed medications for RD (glucocorticoids, csDMARDs, or bDMARDs). 

The final 11,636 cases were cross-checked with the death registry of the health Information center at the Institute of Hygiene, and the date and cause of death were obtained if the death had occurred between 2013 and 2021. The cases were cross-checked using personal identification codes. The principal causes of death were compared between the prepandemic and pandemic time periods, and grouped under the following categories: deaths because of cardiovascular and circulatory causes, malignancies, infections, respiratory causes, musculoskeletal diseases, external causes of death, and unspecified causes of death. COVID-19, as a recorded cause of death, was only calculated in the pandemic period.

The final dataset used for analysis included sex, age, the ICD-10 code of RD, the date of RD diagnosis, the date and cause of death if applicable, and information about the prescription of state-reimbursed drugs.

For the comparison with the general population of Lithuania, information on the adult population census in 2013–2021 was obtained from Statistics Lithuania (www.stat.gov.lt, accessed on 27 November 2022).

### 2.2. Statistical Methods

Sex- and age-standardized mortality ratios (SMRs) were calculated by dividing the observed number of deaths in the RD patients’ cohort by the expected number of deaths, calculated using the national rates from the Lithuanian Department of Statistics’ official statistics website; 95% confidence intervals (CIs) for the SMRs were calculated.

All statistical analyses were performed using Microsoft Excel (2016) by the Microsoft Corporation. 

## 3. Results

### 3.1. Demographic Characteristics of the RD Cohort 

During the period of 2013–2019, 11,636 patients with newly diagnosed RD were identified comprising 6008 patients with RA, 3289 with SPA, and 2339 with systemic CTD and vasculitis. The cohort was further followed up during the COVID-19 pandemic years (2020–2021); the mean duration of total follow-up was 5.49 (standard deviation 2.22) years. The mean age of the patients at the time of RD diagnosis was 57 years (range, 18–97). 52% of the total cohort were RA patients. The majority of the cohort patients were women (70%), particularly in the RA and systemic CTD groups (77% and 76%, respectively). In SPA, the group gender distribution was equal (52% of women and 48% of men).

### 3.2. Death Cases and Leading Causes of Death

At a total follow-up of 63,901.16 person-years, 950 death cases had occurred in 2013–2019, and 581 death cases in 2020–2021. The demographic data of the death cases are presented in Table 1. Around 60% of the death cases were women in both periods of time. The majority of the death cases were observed in the RA group during both periods (54% in the 2013–2019 period and 55% in the period of 2020–2021). Age at the time of death in the RD cohort was higher in the pandemic period (76.42 years) compared with 73.5 years during the 2013–2019 period. SPA patients were the youngest at the time of death, around 67 years in both the analyzed periods of time, whereas the RA and CTD groups did not differ much from the average in either time period.

We compared the main causes of death between the prepandemic and pandemic cohorts of patients. Despite the COVID-19 pandemic and deaths occurring because of this disease, the predominant causes of death for both cohorts were cardiovascular diseases and neoplasms. However, the reported COVID-19 disease was the third most common direct cause of death in the pandemic cohort. We present the proportions of the main causes of death in both cohorts in Table 2.

### 3.3. Death Rates and Standardized Mortality Ratios 

Death rates that were adjusted to the general population were separately calculated in the RD cohort for the prepandemic and pandemic periods and compared to the national death rates. In the prepandemic period of 2013–2019, the adjusted death rate observed in the RD group was higher than that of the general population of Lithuania (2239.00 per 100,000 patient years and 1702.97 per 100,000 inhabitants per year, respectively). The excess of deaths of almost 16% was obvious for the general Lithuanian population during the pandemic period (the death rate in the period of 2020–2021 was 1973.39 per 100,000 inhabitants per year), while it was not the case for the adjusted RD cohort (excess of deaths in pandemic period, −27.2%), meaning that RD patients endured the pandemic period better than the general population did. Death rates are presented in Table 3.

The age- and sex-adjusted SMRs were calculated for the RD cohort for both periods of time. RD mortality was higher than that in the general population in the prepandemic years (the total SMR of the cohort was 1.32 (95% CI 1.23;1.40)), but in the pandemic years, RD mortality was significantly lower than that in the general population (the SMR of the total RD cohort was 0.83 (95% CI 0.76;0.90)). This trend was observed in RA and SPA patients except for the CTD and vasculitis group. The SMR in the CTD and vasculitis group did not differ from that of the general population, at 0.93 (95% CI 080; 1.07)). The SMRs of the total RD cohort and different RDs are presented in Table 4. 

## 4. Discussion

In this article, we describe the mortality results of the follow-up of a large Lithuanian RD patient cohort starting from the diagnoses and following to the end of 2021. We found that the COVID-19 pandemic did not influence the mortality of the cohort. On the contrary, RD patients survived the pandemic period better than the general Lithuanian population did. The age- and sex-adjusted SMR of the total RD cohort was 0.83 (compared to an SMR of 1.32 in the prepandemic years). This trend was observed in the RA and SPA patient groups. Only mortality in the CTD and vasculitis group was a little higher, but did not differ from that of the general population, with an SMR of 0.93 (95% CI 0.80; 1.07).

The COVID-19 pandemic has been generally linked to an increased global mortality and excess of deaths. At the very beginning of the pandemic, patients diagnosed with cardiovascular disease (CVD), diabetes, chronic respiratory disease, hypertension, cancer, and older age had an increased mortality risk [15]. In 2021, Dessie et al. performed a systematic review and meta-analysis of the risk factors of COVID-19-related mortality. The meta-analysis of more than 400,000 cases of COVID-19 concluded that the male sex, older age, current smoking status, obesity, increased D-dimer level, and comorbidities such as acute kidney injury, chronic obstructive pulmonary disease, diabetes, hypertension, CVD, and malignancy determined a higher risk of death [16]. Predictors of the increase in all-cause mortality during the COVID-19 pandemic were also analyzed, revealing that people who were aged 65–79 years, single, and had elementary-school education (or below) were at higher risk for excess death from any cause [17]. In addition, a study from the USA revealed racial and ethnic disparities, reporting a twice higher rate of excess deaths in Black, American Indian/Alaska Native, and Latino males and females compared with White and Asian people [18]. Carey et al. reported that not only older age (>80 years) and non-White ethnicity are risk factors for excess mortality, but also a high body mass index (>40), dementia, learning disabilities, severe mental illnesses, and the specific place of residence (care home, most deprived) [19]. 

There are global data that the COVID-19 pandemic had a definite effect on different aspects of inflammatory RDs due to the challenges to healthcare systems, shortages of resources, limitations in performing routine care, possible delays in diagnosis, and even, in some cases, supply gaps of some medications, such as hydroxychloroquine [20]. Although it seems that there is no evidence that rheumatic patients are at higher risk of contracting SARS-CoV-2 [21], it is yet not clear whether patients with inflammatory RD are at a greater risk of COVID-19-related mortality. A number of studies have addressed this problem so far. A study conducted in South Korea examined data from 8297 patients diagnosed with inflammatory RD and reported an increased mortality rate among the aforementioned patients during the pandemic. They also found that taking ≥10 mg of systemic glucocorticosteroids was associated with a higher risk of testing positive for COVID-19, developing severe illness, and COVID-19-related death [22]. A Greek study that covered 12-month data derived from electronic databases of around 11 million Greek residents indicated that mortality due to COVID-19 was higher in patients with systemic sclerosis and RA in comparison to that of the general population, while the mortality rates in ankylosing spondylitis, systemic lupus erythematosus, and PsA were equal to those of the comparators [23]. A Swedish nationwide study also demonstrated that mortality rates differred among different types of arthritis. For example, RA has higher mortality rates; in the SPA group, mortality levels matched those of the general population [24]. A nationwide cohort study from Denmark also matched the Swedish data: patients diagnosed with RA were more likely to have a severe outcome (intensive care unit admission, acute distress syndrome, or death) of COVID-19 [25]. However, data throughout the literature are inconsistent. For example, an American multicenter study demonstrated that mortality rates in the general population and in patients with inflammatory RD were equal [26]. 

There is a discussion regarding which factors could have added to excess mortality during the COVID-19 pandemic. Apart from the lethal outcomes directly caused by COVID-19 infection, deaths caused by the collapse or overstretching of medical systems due to the COVID-19 pandemic may have also added to the excess deaths [27,28] There is recorded evidence about the higher rates of anxiety and depression during the pandemic period, which might have led to an increase in deaths from suicide, as was estimated in Japan [29,30,31]. 

There are also generally accepted factors that lowered excess mortality. Isolation requirements and other pandemic restrictions might have decreased the incidence of some conditions and injuries, such as traffic accidents, thereby resulting in a decrease in death rates due to these causes [32,33]. The lower rate of deaths from chronic respiratory conditions could have been influenced by the reduction in air pollution [34]. There was also noted decrease in deaths from chronic conditions such as ischemic heart disease or chronic respiratory disease, probably because these frail individuals were at higher risk to die from COVID-19 rather than from these chronic conditions in the pandemic period [35]. However, the most likely reason for the non-COVID-19 mortality to have decreased is the lower incidence of influenza and other infectious respiratory diseases during the COVID-19 pandemic—a decrease in cases of 80% or more was reported by the WHO. It was estimated that the decrease in influenza incidence alone could have led to a decrease in total annual mortality of 3–6% [13]. The main reasons for the decrease in the rate of respiratory infections are behavioral factors—mask use, reduced mobility, social distancing, and lockdown measures, which were really strict in Lithuania during the most severe pandemic periods. Lastly, the use of SARS-COV-2 vaccines has indisputably considerably lowered mortality rates among people who contracted the virus and among the general population. Inflammatory RD patients were among those who were the first to be vaccinated in Lithuania—as a group at higher risk, their vaccination began as early as April 2021. The strict lockdown measures and social distancing imposed by the state authorities, and by the conscious decision of immunocompromised patients themselves, as well as early vaccination were the main factors that resulted in the reduced mortality in the Lithuanian RD cohort during the pandemic years.

The main asset of our study is the populational coverage. We obtained data on the patients with RD and their death cases from the entire Lithuanian population from official state-run sources. No death case could have been omitted due to this. 

The main limitation of the study is the possible exclusion of some RD cases if they had not been treated with the medications reimbursed by the state, as this was one of the exclusion criteria for the verification of the cases. This could have resulted in the exclusion of mild cases, and especially SPA patients, because the treatment of SPA is currently poorly reimbursed in Lithuania. Another limitation is that we followed up the cohort of the patients first diagnosed with RD during the period of 2013–2019 and did not include newly diagnosed RD cases during the pandemic years. COVID-19 infection itself could be a trigger of RD. A notable proportion of patients with COVID-19 present with fatigue, muscle pain, and other rheumatic manifestations, such as arthralgia, vasculitis, and chilblains, and up to 49% of COVID-19 patients present with different autoantibodies (antinuclear, antiphospholipid) [36,37]. However, it is not clear whether these antibodies show a temporary immune dysregulation or whether SARS-CoV-2 is capable of causing RD [38,39,40]. A number of new-onset rheumatic autoimmune diseases following SARS-CoV-2 infection were reported, with the most common being systemic vasculitis and inflammatory arthritis [41]. A few cases of inflammatory myopathies [42], systemic lupus erythematosus [43], and systemic sclerosis [44], during or after COVID-19, were also reported. However, it is unclear whether SARS-CoV-2 infection uncovers and accelerates previously subclinical rheumatic illness or induces de novo disease, since no direct causal relationship has been proven. On the other hand, an immense study in the USA demonstrated only a small rate of new-onset RD in patients with a positive polymerase chain reaction (PCR) test for the detection of SARS-CoV-2 (6 incident cases among over 15,200 patients), which was similar to the rate among the matched controls with a negative PCR test (five incident cases) [45]. However, this frequency might have been lower due to steroid therapy in COVID-19 pneumonia [46]. Other authors found that post-COVID-19 was strongly associated with the erythrocyte sedimentation rate and C-reactive protein, but not autoantibodies, which suggested an inflammatory rather than autoimmune mechanism of arthritis [47]. This could also explain the significantly increased number of new Kawasaki disease cases during the COVID-19 pandemic [48]. Derksen et al. compared the clinical phenotype and autoantibody patterns in patients with polyarthritis after COVID-19 and in RA patients, and concluded that RA following COVID-19 was seemingly a coincidence [49]. Taking the aforementioned into account, evidence on the role of SARS-CoV-2 in inducing RD is currently unclear, and large-sample studies with an adequate follow-up are still needed. 

## 5. Conclusions

In conclusion, the data from existing studies referring to the mortality of patients with RD are inconsistent, and our study added new evidence that the RD patient mortality rate was less than that of the general population. We speculate that the proper management of the pandemic in the country and in the RD patient group resulted in a lower mortality rate than what was expected. The relation between COVID-19 and new occurrences of RD in Lithuanian cohort needs to be explored case by case and in a longer follow-up.

## Figures and Tables

**Table 1 medicina-59-00311-t001:** Demographic characteristics of rheumatic-disease patient cohort’s death cases in the prepandemic and pandemic periods.

Deaths/Periods	Prepandemic Period (2013–2019)	Pandemic Period (2020–2021)
Total number of deaths	950	581
Female (%)	562 (59.16)	358 (61.62)
Male (%)	388 (40.84)	223 (38.38)
Number of deaths in different disease groups:		
RA group (%) *	509 (53.58)	321 (55.25)
SPA group (%) **	142 (14.95)	81 (13.94)
CTD and vasculitis group (%) ***	299 (30.63)	179 (30.81)
Age at the time of death:		
Mean total-cohort age (SD ****)	73.50 (12.33)	76.42 (11.35)
Mean RA group age (SD)	74.95 (11.45)	77.64 (9.95)
Mean SPA group age (SD)	67.34 (13.45)	67.96 (14.75)
Mean CTD and vasculitis group age (SD)	73.98 (12.37)	78.04 (10.22)

* RA—rheumatoid arthritis, ** SPA—spondyloarthropathy, *** CTD—connective-tissue disease, **** SD—standard deviation.

**Table 2 medicina-59-00311-t002:** Main causes of death in patients with rheumatic diseases in the prepandemic and pandemic cohorts.

Causes of Death	Prepandemic Cohort (2013–2019) *n* = 950	Pandemic Cohort (2020–2021) *n* = 581
Cardiovascular diseases (%)	**450 (47.4) ***	**266 (45.8)**
Neoplasms including lymphopoietic system (%)	**220 (23.2)**	**103 (17.7)**
Diseases of the respiratory system (%)	57 (6.0)	15 (2.6)
Diseases of the musculoskeletal system (%)	48 (5.1)	15 (2.6)
External causes of death (%)	38 (4.0)	15 (2.6)
Other causes of death (%)	**137 (14.4)**	81 (13.9)
COVID-19 (%)	0	**86 (14.8)**

* The three most common causes of death are represented by numbers in bold.

**Table 3 medicina-59-00311-t003:** Death rates and excess of deaths observed in the general population of Lithuania and the cohort of rheumatic diseases during the prepandemic and pandemic periods.

	Prepandemic Period (2013–2019)	Pandemic Period (2020–2021)	Excess of Deaths
General population of Lithuania (average number of deaths per 100,000 inhabitants per year)	1702.97	1973.39	15.9%
RD * cohort (death rate, standardized according to age and sex, 95% CI)	2239 (2099;2386)	1630 (1500;1768)	−27.2%

* RD—rheumatic diseases.

**Table 4 medicina-59-00311-t004:** Standardized mortality ratios in the RD cohort during the prepandemic and pandemic periods.

	Standardized Mortality Ratios in Prepandemic Period (95% CI)	Standardized Mortality Ratios in Pandemic Period (95% CI)
Total:	1.32 (1.23;1.40)	0.83 (0.76;0.90)
Women	1.31 (1.21;1.42)	0.79 (0.71;0.87)
Men	1.32 (1.19;1.46)	0.90 (0.78;1.02)
RA *	1.25 (1.14;1.36)	0.83 (0.74;0.92)
SPA **	1.16 (0.98;1.37)	0.67 (0.53;0.83)
CTD and vasculitis ***	1.55 (1.38;1.73)	0.93 (0.80;1.07)

* RA—rheumatoid arthritis, ** SPA—spondyloarthropathy, *** CTD—connective tissue diseases.

## Data Availability

The data presented in this study are available on request from the corresponding author. The data are not publicly available because they contain the personal identification codes of patients included in this study; therefore, they are sensitive personal data that are strictly protected by local ethical restrictions and the rules of personal data safety in Lithuania.

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
