# Peer review of "How Did the Two Years of the COVID-19 Pandemic Affect the Outcomes of the Patients with Inflammatory Rheumatic Diseases in Lithuania?"

_medicina, 2023, doi:10.3390/medicina59020311_

Round 1

Reviewer 1 Report

The present work was aimed at reporting data from a retrospective study, in which the authors identified patients with the first-time diagnosis of inflammatory RD and followed them up during the pandemic years 2020-2021. Death rates and sex- and age- 19 standardized mortality ratios (SMR) were calculated for the pre-pandemic and pandemic periods 20 of time. Data from 11 636 patients newly diagnosed with inflammatory RD were obtained, followed-up until the end of 2021 or the fact of death. The median duration of the follow- 22 up was 5.5 years. 1531 cases of death occurred between 2013 and 2021. The prevailing causes of 23 death in pre-pandemic period were cardiovascular diseases, neoplasms and diseases of respiratory 24 system. In the pandemic years cardiovascular diseases and neoplasms remained the two most often 25 causes of death, with the reported COVID-19 disease in the third place. SMR of the total RD cohort 26 was calculated to be 0.83. On the basis of the results, the authors concluded that COVID-19 pandemic did not influence the mortality of the RD patients. Data are well reported, even if they are in contrast to what already discussed in the literature, however important. 

I have some comments:

-       Although the scope of the work is to study mortality, it is important also to discuss more in detail the potential role of therapy (or no adherence to therapy). Or, in case of lacking data about that, to discuss more in detail the potential role also of the disease activity on mortality.

-       For sake of the completeness,I would suggest mention the following paper coming from a big network on the impact that the pandemic had on clinical and organizational aspects of rheumatic patients: Talarico R, Aguilera S, Alexander T, Amoura Z, Antunes AM, Arnaud L, Avcin T, Beretta L, Bombardieri S, Burmester GR, Cannizzo S, Cavagna L, Chaigne B, Cornet A, Costedoat-Chalumeau N, Doria A, Ferraris A, Fischer-Betz R, Fonseca JE, Frank C, Gaglioti A, Galetti I, Grunert J, Guimarães V, Hachulla E, Houssiau F, Iaccarino L, Krieg T, Limper M, Malfait F, Mariette X, Marinello D, Martin T, Matthews L, Matucci-Cerinic M, Meyer A, Montecucco C, Mouthon L, Müller-Ladner U, Rednic S, Romão VC, Schneider M, Smith V, Sulli A, Tamirou F, Taruscio D, Taulaigo AV, Terol E, Tincani A, Ticciati S, Turchetti G, van Hagen PM, van Laar JM, Vieira A, de Vries-Bouwstra JK, Cutolo M, Mosca M. The impact of COVID-19 on rare and complex connective tissue diseases: the experience of ERN ReCONNET. Nat Rev Rheumatol. 2021 Mar;17(3):177-184. doi: 10.1038/s41584-020-00565-z. Epub 2021 Jan 6. Erratum in: Nat Rev Rheumatol. 2022 Dec;18(12):734. PMID: 33408338; PMCID: PMC7786339.

Reviewer 2 Report

Abstract

1.     the study is retrospective

2.     The median duration of the follow-up was 5.5 (?) years. (5.5 ± (?) should be in the form).

Materyals and Metods

1.     material - method should be written more clearly

2.     PsA and AS are in the SpA group. Why were these two diseases evaluated separately from the SpA group?

3.     Which of the biosimilar drugs ?

4.     Flow chart can be added

Statistical methods

1.       We have conducted a retrospective cohort study to assess the mortality in the study 113 population and compare rates between the pre-pandemic period (2013-2019) and COVID- 114 19 pandemic years (2020-2021). (this sentence is mentioned in the material - method, no need).

2.       The duration of follow-up was calculated from the date of RD diagnosis to the first of one of the following events: death or the end of follow-up (December 31, 2021). (no need).

3.       These were compared between pre-pandemic and pandemic time periods and grouped under the following categories: deaths because of cardiovascular and circulatory causes, malignancies, infections, respiratory causes, musculoskeletal diseases, external causes of death and unspecified cause of death  (this part should be in the material-method section).

4.       Which statistical method was used, Which statistical package was used (which version …etc)

must be written (Statistical method should be re-evaluated).

Results

1.       3289 with SPA and PsA, (should be SpA only (not PsA) (written correctly in table 1)

2.       SD description should be written under table 1

3.       in table 2, is the pandemic cohort 102%?

4.       Long names should be written for the abbreviations used in Table 4.
